# Differential Gene Expression in the Upper Respiratory Tract following Acute COVID-19 Infection in Ambulatory Patients That Develop Long COVID

**DOI:** 10.3390/pathogens13060510

**Published:** 2024-06-17

**Authors:** Mia J. Biondi, Mary Addo, Muhammad Atif Zahoor, Elsa Salvant, Paul Yip, Bethany Barber, David Smookler, Sumaiyah Wasif, Kayla Gaete, Christopher Kandel, Jordan J. Feld, Hubert Tsui, Robert A. Kozak

**Affiliations:** 1School of Nursing, York University, Toronto, ON M3J 1P3, Canada; 2Toronto Centre for Liver Disease, University Health Network, Toronto, ON M5G 2C4, Canada; atif.zahoor@uhn.ca (M.A.Z.); bethany.barber@uhn.ca (B.B.); david@vircan.ca (D.S.); jordan.feld@uhn.ca (J.J.F.); 3Department of Laboratory Medicine and Pathobiology, University of Toronto, Toronto, ON M5S 1A8, Canada; mary.addo@mail.utoronto.ca (M.A.); paul.yip@sunnybrook.ca (P.Y.); hubert.tsui@sunnybrook.ca (H.T.); 4Sunnybrook Health Sciences Centre, Sunnybrook Research Institute, Toronto, ON M4N 3M5, Canada; elsa.salvant@utoronto.ca (E.S.); kayla.gaete@sri.utoronto.ca (K.G.); 5Precision Diagnostics and Therapeutics Program, Department of Laboratory Medicine and Molecular Diagnostics, Sunnybrook Health Sciences Centre, Toronto, ON M4N 3M5, Canada; 6Michael Garron Hospital, Toronto, ON M4C 3E7, Canada; christopher.kandelmd@tehn.ca; 7Department of Immunology, University of Toronto, Toronto, ON M5S 1A8, Canada

**Keywords:** long COVID, differential gene expression, viral clearance, pathway dysregulation

## Abstract

**Background:** Post-acute sequelae of COVID-19, or long COVID, is a condition characterized by persistent COVID-19 symptoms. As long COVID is defined by clinical criteria after an elapsed period, an opportunity for early intervention may aid in future prophylactic approaches; however, at present, the pathobiological mechanisms are multifactorial. By analyzing early virally infected upper respiratory tract tissue prior to eventual clinical diagnosis, it may be possible to identify biomarkers of altered immune response to facilitate future studies and interventions. **Methods:** This is a sub-group analysis of samples collected from those with confirmed COVID-19. RNA extraction from nasopharyngeal/mid-turbinate samples, sequencing, and bioinformatic analysis were performed to analyze long COVID and non-long COVID cohorts at day 14 post infection. Differences in mean viral load at various timepoints were analyzed as well as serological data. **Results:** We identified 26 upregulated genes in patients experiencing long COVID. Dysregulated pathways including complement and fibrinolysis pathways and IL-7 upregulation. Additionally, genes involved in neurotransmission were dysregulated, and the long COVID group had a significantly higher viral load and slower viral clearance. **Conclusions:** Uncovering early gene pathway abnormalities associated with eventual long COVID diagnosis may aid in early identification. We show that, post acute infection, in situ pathogenic deviations in viral response are associated with patients destined to meet consensus long COVID diagnosis that is entirely dependent on clinical factors. These results identify an important biological temporal window in the natural history of COVID-19 infection and long COVID pathogenesis amenable to testing from standard-of-care upper respiratory tract specimens.

## 1. Introduction

The complications of SARS-CoV-2 infection remain incompletely understood, and as COVID-19 cases increase, so does the number of individuals afflicted with a myriad of ongoing symptoms long after the acute phase of the disease. This post-viral syndrome can be debilitating and greatly affect quality of life. This condition, referred to as post-acute sequelae of SARS-CoV-2, or long COVID (LC), affects an estimated 10–40% of individuals and can be influenced by a number of factors including vaccine doses and infecting strain [1]. A recent examination of 138,818 individuals with COVID-19 from the United States Veteran Affairs indicated that the risk of LC-associated symptoms remained higher even at the 2-year mark compared to uninfected controls [2].

The pathogenesis of LC is not well understood either, and studies have suggested that reinfection, vaccination, and viral variants likely play a role in whether an individual develops LC [3,4]. A study of 309 COVID-19 patients identified auto-antibodies, type 2 diabetes, SARS-CoV-2 RNAemia, and Epstein–Barr virus viremia as risk factors for developing LC [5]. Additionally, recent work by Wong and colleagues has shown that a depletion of serotonin levels is seen in LC patients and may be the result of TLR3 activation by viral RNA causing ongoing type I interferon production [6]. The dysregulation of the innate immune response in LC was also supported by recent findings that complement activation and thrombo-inflammation are more prevalent in these patients [7].

RNA-sequencing is a useful tool for investigating the host response to pathogens and has been used to characterize the innate immune response to SARS-CoV-2 infection and identify prognostic gene signatures for COVID-19 outcomes [8,9]. In addition, this tool allows the identification of novel features with a dynamic range. As such, it allows for the host response to be evaluated at different timepoints and conditions associated with disease states, and can provide insight into pathogenesis. Recently, the RNA-sequencing of peripheral blood mononuclear cells identified that pathways associated with cell death and fibrosis are upregulated in LC patients [10]. At present, the host response underlying LC is not fully understood, specifically during the acute phase of the disease, which may skew the immune system towards ongoing inflammation [11]. Given its impact on quality of life, investigating the underlying mechanisms is an important endeavor, leading to improved disease recognition and potential interventions. Additionally, the identification of potential early biomarkers is important, as this could allow for a more biologically homogeneous classification of patients that is currently entirely dependent on subjective clinical criteria.

Here, we investigated the host gene signatures present from the upper respiratory mucosa at the end of the acute phase of infection between individuals who subsequently developed LC, as currently defined, and those who did not. By associating the host transcriptional and immunological response in the upper respiratory tract with the subsequent development of long COVID, as determined by an established scoring system [12], we highlighted unique gene expression patterns in these individuals. Additionally, we examined the humoral immune response in this group. These findings highlight the potential novel applicability of RNAseq from these samples to predict future long COVID.

## 2. Materials and Methods

### 2.1. Cohort Description

Residual samples from a subgroup of participants enrolled in the placebo arm of two double-blind, placebo-controlled outpatient trials of peg-interferon lambda as a treatment for COVID-19 were evaluated [13,14]. The first trial was conducted between May and September 2020, and recruited from six institutions in Toronto, Canada. Participants were eligible if within 7 days of symptom onset or first positive molecular test if initially asymptomatic, and between the ages of 18 and 70. Exclusion criteria were pregnancy and medical conditions that could be worsened by the study drug interferon-lambda [13]. The second study was conducted from September 2021 to March 2022, as part of an international phase III double-blind, placebo-controlled COVID-19 outpatient treatment trial. Individuals recruited were symptomatic COVID-19 patients aged 18 or older, who had at least one risk factor for severe disease [14]. Only samples collected from the Toronto, Canada, sites were used. All institutions involved locally approved the study, Health Canada approval was received, and the trials were registered at clinicaltrials.gov (NCT04354259 and NCT04727424). From the trial cohort, only samples from patients in the placebo arms of each trial were included and all samples were positive by RT-PCR using our assay that has been previously described [13]. Residual swab material was available from 27 participants between both trials from specimens collected at study enrolment, and at the end of the acute phase of illness (d = 14) for RNAseq analysis. At the follow-up visit, during the convalescent phase (day 90 or later), individuals were evaluated by the study team and a questionnaire to evaluate symptom number and severity was performed. Individuals were determined to have LC based on the composite symptom score developed by Thaweethai et al. [12]. All patients characterized as LC had a symptom score of 12 or greater.

### 2.2. Nucleic Acid Extraction and Viral Load Quantification

Total nucleic acid extraction from midturbinate (MT) or nasopharyngeal (NP) swabs was performed using the NucliSENS EasyMAG (bioMérieux, Saint-Laurent, QC, Canada) or QIAamp viral RNA minikit (Qiagen, Toronto, ON, Canada), and the detection of SARS-CoV-2 RNA was performed using established clinical assays [13]. RNA was stored at −80 °C for analysis. The total nucleic acid was separated on a 2100 Bioanalyzer system using the RNA 6000 Pico Kit (Agilent, Santa Clara, CA, USA) for sample quality assessment. The amount of amplifiable human origin total RNA was estimated by amplifying exons spanning the 96 bp coding region in the human β-glucuronidase gene (GUSB). The qPCR reaction was carried out in 10 μL with 1 × TaqMan Fast Virus Mix, 1 × Human GUSB assay mix (Hs0093962_m1, ThermoFisher Scientific, Waltham, MA, USA), and 2 μL of extracted total RNA using the StepOnePlus Real-Time PCR System (ThermoFisher Scientific). The PCR cycling conditions were 50 °C for 5 min, 95 °C for 20 s, 40 cycles of 95 °C for 3 s, and 60 °C for 1 min. Standard curves for total RNA quantification were prepared using HL-60 Total RNA (ThermoFisher Scientific).

### 2.3. RNA Sequencing

Aliquots from MT/NP swabs were sequenced using the Ion Torrent platform comprising a single multiplexed panel targeting 18,574 coding and 2228 non-coding RefSeq genes (>95% of the RefSeq gene database). Targeted human transcriptome sequencing was performed on the Ion S5XL Next Generation Sequencing system with the Ion AmpliSeq Transcriptome Human Gene Expression Panel (ThermoFisher Scientific). RNA was quantified using a GUSB qPCR assay and treated with ezDNase for the removal of genomic DNA contamination in the sample, and then reverse-transcribed in a 15-µL reaction using the SuperScript IV VILO Master Mix with an ezDNase Enzyme kit. The barcoded complementary DNA libraries were constructed using an Ion Ampliseq Transcriptome Human Gene Expression Panel, Chef-Ready Kit. The targeted sequences were PCR-amplified for 18 cycles with a 16 min extension time. The quantification of the libraries was performed using the Ion Library TaqMan Quantitation Kit (ThermoFisher Scientific). Sequencing template preparation was performed using Ion Chef with Ion 550 Chef Kits. Sequencing was performed for 500 flows on an Ion S5XL Sequencer with an Ion 550 chip.

### 2.4. Serology

Serum was collected from participants at the end of the acute phase of illness (day 14) and during the convalescent phase, (greater than 3 months after diagnosis) which is associated with long COVID (day > 90). Serum underwent analysis to quantify anti-Spike and anti-Nucleocapsid IgG titers using the Roche Elecsys immunoassay, using previously described methods [15].

### 2.5. Bioinformatics Analysis

Sequence data were analyzed using the Galaxy web platform [16]. Raw reads were trimmed using Trimmomatic (Galaxy Version 0.38.1) with a required quality of 20 and a window size of 4 bp. Raw reads were mapped to the NCBI genome reference build [17], GRCh38.p14, performed using HISAT2 (Galaxy Version 2.2.1 + galaxy1) with default parameters. Data were surveyed for quality assessment using FastQC (Galaxy Version 0.74 + galaxy0) ensuring per base sequence quality. Subsequent quantification was performed with featureCounts (Galaxy Version 2.0.3 + galaxy1) using the NCBI hg38 annotation GTF [17] with default parameters. Differential gene expression (DE) was analyzed using EdgeR (Galaxy Version 3.36.0 + galaxy2) using the Trimmed Mean of M-values normalization method to account for variation among reads. Normalized count data were used for further visualization of gene expression. Functional analysis was performed initially using Enrichr [18,19,20] to identify significant pathways shared among various software platforms using a gene list of statistically significant DEs. Gene sets of interest and the downstream analysis of gene interactions were analyzed through STRING (Version 12.0) [21].

### 2.6. Statistical Analysis

Low-expressed genes were filtered out with a minimum CPM of 1.0 on minimum samples of 0. A minimum log-fold change of 1.5 and an adjusted *p*-value of less than 0.05 were used. The Benjamin–Hochberg (BH) false discovery technique was used as a correction method for EdgeR-adjusted *p*-values to assess significance. The Gene Ontology of protein–protein interactions identified through STRING with an FDR of less than 0.05 was considered significant. Viral load data were analyzed for significance using unpaired *t*-tests. A parametric test was used (assuming Gaussian distribution), with a *p*-value of less than 0.05 being considered significant. A Kaplan–Meier survival analysis was performed on the viral clearance data. The logrank (Mantel-Cox test) was used to compare viral loads and the Gehan–Breslow–Wilcoxon test was used to compare viral RNA clearance between the two cohorts (*p* < 0.05).

## 3. Results

### 3.1. Cohort Description

A total of 45 patients were selected for analysis, among whom we identified 34 non-LC and 11 LC patients. All individuals were positive for SARS-CoV-2 by RT-PCR in their samples collected at the initial time point. From this, 18 non-LC and 9 LC participants were included for RNAseq analysis due to the availability of residual clinical material. Demographic details are provided in Table 1. In the LC group, the mean symptom score was 15, and in the non-LC group, the mean score was 1.8. No individuals reported a reinfection during the time between their day-14 visit and their subsequent follow-up visit after 90 days.

### 3.2. Quantification of Viral RNA in Respiratory Specimens

Viral RNA persistence in the nasal swabs from LC and non-LC patients was evaluated. In the non-LC cohort, consisting of 18 patients, 81% of patients were RNA-positive at enrollment, 33% at day 7, and 6% at day 14. In the LC group of 9 patients, 75% of patients were RNA-positive at enrollment, 77% at day 7, and 22% at day 14. Lower viral loads were observed in the non-LC group throughout the course of the study. The difference in mean viral load showed significance at days 1 and 10 between the two groups (Figure 1).

### 3.3. Distinct Gene Expression Patterns and Pathways Associated with Long COVID

The early host responses of LC and non-LC patients were compared from samples collected at enrollment and day 14 to identify uniquely dysregulated genes during the acute phase of infection. To accurately identify which individuals were classified as long COVID patients, differential gene expression was analyzed for cohorts defined using two different LC definitions: the World Health Organization (WHO) LC definition and criteria [22] and the LC scoring criteria developed by Thaweethai et al. [12]. No significantly differentially expressed genes were identified at the enrollment time point. Using the Thaweethai classification system, gene expression patterns were compared from samples at the end of acute illness (Figure 2a) and twenty-six genes were found to be upregulated (Table 2). When patient samples were analyzed based on the WHO criteria, fewer differentially expressed genes were identified (Figure 2b), suggesting that the Thaweethai classification produced a larger analysis background. Unique genes identified using the WHO criteria included SLC6A2, a known neurotransmitter gene. Common enriched genes in the LC cohort between both scoring methods included YWHAE, KLK4, LCE2C, KRT79, and RPS6KA1. Interaction networks for these genes were analyzed through STRING to identify significant biological processes and pathways. YWHAE encodes the 14-3-3 epsilon regulator protein and is a target of SARS-CoV-2. The gene was enriched for the negative regulation of dendritic cell apoptotic processes and the negative regulation of synaptic vesicle exocytosis. LCE2C is known to be involved in defence against Gram-positive bacteria. Given the upregulation of the gene, this may relate to LC patients being more susceptible to secondary bacterial infections post COVID [23]. We noted that 77% (*n* = 7/9) of LC patients experienced brain fog and were therefore compared to the 18 non-LC patients in a separate analysis in which the top thirty significant DEGs were identified (Figure 2c). This was done to determine if brain fog is indicative of differences in gene expression. Although most patients in the LC group experienced this symptom, it may not be a key factor as no significant changes were found here or through functional analysis.

### 3.4. Dysregulated Pathways in LC Patients

Pathway analysis was performed on patient samples classified using the criteria developed by Thaweethai et al. [12]. Enriched pathways identified with EnrichR (*p* < 0.05) included the classical complement pathway and the fibrinolysis pathway. In addition, IL-7 signal transduction was identified with IL-7 signaling pathways including the regulation of T cell proliferation, memory T cell survival, and B cell development (Table 3).

### 3.5. Serology

Clinically assayed anti-S serological data were available for 29/45 individuals at day 14 and 35/45 individuals at day 90. Anti-S titers in the LC and non-LC cohorts appeared relatively equal by day 90, with a difference of means ± standard error of the mean between timepoints of 2867 ± 2692. Anti-N data were available for 29/45 individuals at day 14 and 36/45 at day 90. Most patients were anti-N-reactive by day 90. Overall, there were no significant differences identified between cohorts in both anti-S and anti-N at both timepoints (Figure 3). This suggests that clinical serology may not be able to distinguish LC from non-LC patients.

## 4. Discussion

Long COVID is a public health burden that needs further investigation to robustly characterize the condition. Our study identified gene expression patterns and pathways that differed between LC and non-LC patients, which reflect the mucosal response at the end of the acute infectious phase. There was a clear trend in viral load, indicating that LC patients have a greater mean viral load over time. In addition, time to viral clearance was significantly prolonged in LC patients. Interestingly, in contrast to the study by Su et al. [5], none of the individuals in the LC group had detectable SARS-CoV-2 in the samples taken during the convalescent phase. Several common DEGs were found through both scoring methods, although a greater number of statistically significant DEGs were identified with the recent LC scoring method. This suggests that the new system, developed by Thaweethai et al. [12], is more reliable in characterizing LC patients as it provides a significantly increased analysis background.

YWHAE expression was increased in the LC cohort. This gene is responsible for ensuring neuronal survival and is targeted by SARS-CoV-2 following infection [25]. The abnormal expression of YWHAE at the end of the acute phase of infection suggests potential early neuronal dysfunction potentially linked to clinical and subjectively reported cognitive impairment. Another upregulated gene, CPB2, plays a role in inhibiting fibrinolysis, and CPB2 and C8G are involved in the regulation of complement cascade. Concordantly, sustained inflammatory response and coagulopathy are known characteristics of COVID-19 infection. Recently, a study identified ongoing thrombo-inflammation in a cohort of LC patients one year after initial diagnosis [7]. In our cohort, we noted an increased expression of the enzyme carboxypeptidase U (CPB2), which may account for the presence of dense clots seen in LC patients as the enzyme is a potent inhibitor of fibrinolysis [26]. Additionally, CPB2 plays a role in the activation of the complement pathway. The upregulation of the complement proteins, CPB2 and C8G, indicates a thrombo-inflammatory response and the over-activation of the immune response. This was displayed in one study in which persistent inflammation in more than half of their cohort of LC individuals was identified [7]. Dysregulation in these complement biomarkers may be indicative of the development of LC progression as their expression is sustained until the end of the acute phase.

The enrichment of IL-7 signalling was identified through pathway analysis and may be an early indicator of LC progression. Increased levels of the pro-inflammatory cytokine have been previously identified in patients with severe COVID-19 symptoms. Furthermore, an increase in Th17 signalling has been seen in COVID-19 patients with severe infection where increased serum IL-7 signalling was associated with increased T cell depletion [27]. ITGA2B was the primary gene found to be involved in IL-7 signal transduction, indicating a Th17 response consistent with the literature. The STRING analysis of ITGA2B gene interactions displayed enrichment for the positive regulation of platelet activation and blood coagulation and the negative regulation of plasminogen activation. This may explain increased coagulation in LC patients because of a lack of fibrin clot lysis.

Many upregulated genes and enriched pathways have been previously identified. Coagulation and complement pathways and interactors were enriched and expected in the LC cohort. In addition to SLC6A2 upregulation, SNAP25 was identified as an enriched gene in the LC cohort, which further suggested disrupted neuro-function and transmission. Inflammatory cytokine responses present during the acute infection phase are known to affect dopamine transmission. This can impact downstream functions including cognition and motor control relating to LC symptoms such as brain fog and abnormal movements. Although neurotransmitter dysregulation was not significantly enriched through initial pathway analysis, the abnormal expression of SLC6A2 and SNAP25 in LC patients suggests dysfunctional neurotransmitter expression. Impaired neurotransmitter release in LC patients has been shown in various studies such as reduced serotonin levels in the post-acute phase [6]. The upregulation of these genes in the LC cohort at day 14 indicates that neurotransmission dysregulation may develop at earlier timepoints during infection.

Serological data from LC and non-LC patients displayed a similar level of anti-N seropositivity at the day > 90 timepoint. No significant trends were found for the anti-S serological data, and mean concentrations in both LC and non-LC groups remained relatively equal at both timepoints. Overall, this suggests that serology may be of little additional benefit as a predictor of long COVID. These data do not account for serological avidity, which may provide better insight into the differences in immune responses between LC and non-LC patients.

There are several limitations to our study. These include the relatively small sample size and the limited number of timepoints where surplus sample was available for RNAseq analysis. The incorporation of more timepoints would be beneficial in producing a broader viewer in gene expression changes throughout the duration of infection. In addition, SARS-CoV-2 strain data were not analyzed in this study and may have influenced gene expression patterns seen among patients. Moreover, the roles of vaccination and infection with different SARS-CoV-2 variants in altering host gene expression patterns in the context of LC need to be more fully examined.

Although our study cohort is relatively small, it should be noted that an extensive questionnaire was performed at pre-specified timepoints to evaluate the clinical presence of LC, and that these patients represent patients from 2020 (pre-vaccination era) to 2022. Additionally, bulk sequencing was performed instead of single-cell sequencing given the use of clinical surplus. However, we believe that the bulk sequencing of the upper respiratory tract may be feasible from a practical, translational perspective given that these results were generated from residual material patient samples readily amenable to incorporation into the clinical workflows.

## 5. Conclusions

Despite its significant morbidity and impact on quality of life, LC requires formal diagnosis after 3 months of symptoms. Current LC definitions remain highly dependent on subjective assessment and may potentially miss windows of therapeutic intervention. In this study, we provide proof of principle that LC patients exhibit divergent upper respiratory tract host responses as early as 14 days post acute infection. The identification of objective biomarkers could aid in better defining LC patients earlier in their natural history as well as further subclassifying LC presentations by pathobiology vs. our current reliance on subjective and possibly confounded clinical symptoms.

## Figures and Tables

**Figure 1 pathogens-13-00510-f001:**
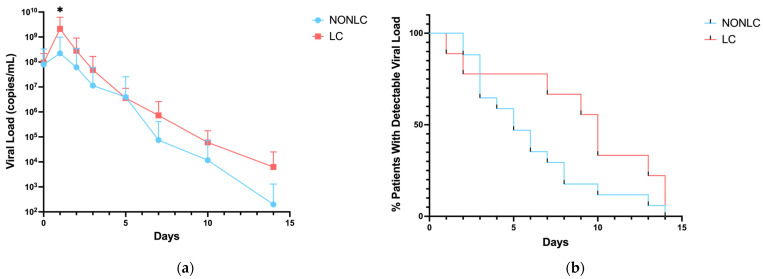
Mean viral load (copies/mL) and clearance. (**a**) The difference in mean viral load among all 45 controls, which included 11 LC and 34 non-LC patients, showed significance at enrollment (*p* = 0.01). (**b**) The viral clearance between all placebo LC and non-LC patients among all controls was not shown to be significant through the curve comparisons. * *p* < 0.05.

**Figure 2 pathogens-13-00510-f002:**
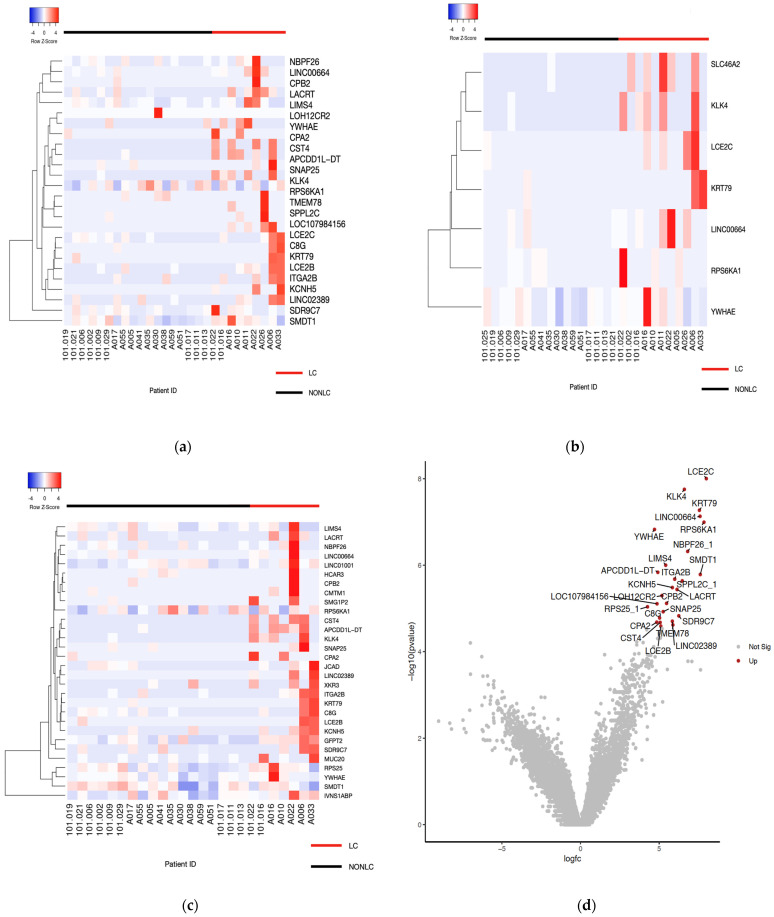
Differential gene expression in LC and non-LC patients. (**a**) Heatmap of day 14 differential gene expression based on criteria from Thaweethai et al. [12], modified from Heatmapper [24]. (**b**) Heatmap of day 14 differential gene expression based on WHO criteria [22], modified from Heatmapper [24]. (**c**) Heatmap of day 14 differential gene expression based on criteria from Thaweethai et al. [12] in LC patients experiencing brain fog compared to non-LC patients. (**d**) Volcano plot of upregulated genes (Galaxy).

**Figure 3 pathogens-13-00510-f003:**
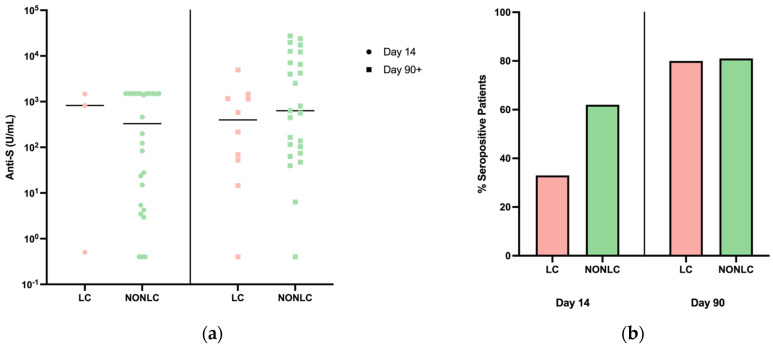
Serology. (**a**) Toronto Together and ILIAD serology results from all placebos at day 14 and 90. Data points display anti-S (U/mL) concentrations of LC and non-LC patients from plasma and/or serum samples. Placebos include patients who were available for sample collection including 3/11 LC and 26/34 non-LC patients at day 14 and 10/11 LC and 25/34 non-LC patients at day > 90. No significance was found. (**b**). Serology results from all individuals at the acute phase (day 14) and the convalescent phase (day > 90). Data points display anti-N (COI) of LC and non-LC patients from plasma and/or serum samples. Samples were from patients who were available for sample collection including 3/11 LC and 26/34 non-LC patients at the acute phase and 10/11 LC and 26/34 non-LC patients at the convalescent phase.

**Table 1 pathogens-13-00510-t001:** Cohort characteristics of individuals who underwent RNAseq.

	LC Patients (*n* = 9)	Non-LC Patients (*n* = 18)
Minimum	29	23
Age, mean age SD	45 ± 10.38	48 ± 14.89
Maximum	56	72
Sex		
Male	2 (22%)	9 (50%)
Female	7 (77%)	9 (50%)
Mean LC Score	15	1.8
Symptoms		
Smell/Taste	8 (88%)	4 (22%)
Post-Exertional Malaise	6 (66%)	2 (11%)
Chronic Cough	2 (22%)	0
Brain Fog	7 (77%)	1 (5%)
Thirst	0	0
Palpitations	4 (44%)	0
Chest Pain	2 (22%)	0
Fatigue	4 (44%)	3 (16%)
Loss of Sexual Desire/Capacity	0	0
Dizziness	3 (33%)	1 (5%)
Gastrointestinal	1 (11%)	0
Abnormal Movements	1 (11%)	0
Hair Loss	0	0

**Table 2 pathogens-13-00510-t002:** Differentially expressed genes.

Gene ID	Gene Name	logFC	logPCM	*p* Value	FDR
**Thaweethai Criteria**					
LCE2C	Late Cornified Envelope 2C	7.99818236	1.02581422	9.96 × 10^−9^	0.00044157
KLK4	Kallikrein-related peptidase 4	6.60280809	−0.4530804	1.77 × 10^−8^	0.00044157
KRT79	Keratin 79	7.55570871	0.45248437	5.37 × 10^−8^	0.0008943
LINC00664	Long Intergenic Non-Protein Coding RNA 664	7.59844381	1.53093465	7.41 × 10^−8^	0.00092692
RPS6KA1	Ribosomal Protein S6 Kinase A1	7.84846419	1.81682067	1.02 × 10^−7^	0.00101709
YWHAE	Tyrosine 3-Monooxygenase/Tryptophan 5-Monooxygenase Activation Protein Epsilon	4.70372922	4.33485505	1.50 × 10^−7^	0.00124748
NBPF26_1	NBPF Member 26	6.81095407	1.81382749	4.81 × 10^−7^	0.00343671
LIMS4	LIM Zinc Finger Domain Containing 4	5.41592334	2.52669888	1.00 × 10^−6^	0.00628014
APCDD1L-DT	APCDD1L Divergent Transcript	4.91514328	−1.3918229	1.47 × 10^−6^	0.00814973
SMDT1	Single-Pass Membrane Protein with Aspartate Rich Tail 1	7.61332542	3.90907796	1.64 × 10^−6^	0.00822243
ITGA2B	Integrin Subunit Alpha 2b	5.98646273	0.29497006	2.11 × 10^−6^	0.00959703
SPPL2C_1	Signal Peptide Peptidase Like 2C	6.46793059	0.27990204	2.32 × 10^−6^	0.0096583
KCNH5	Potassium Voltage-Gated Channel Subfamily H Member 5	5.83849379	−0.4084643	3.31 × 10^−6^	0.01271414
LACRT	Lacritin	6.13395019	1.79756093	3.66 × 10^−6^	0.01305873
LOH12CR2	Loss Of Heterozygosity On Chromosome 12, Region 2	5.17501683	−1.2807788	5.08 × 10^−6^	0.01693806
CPB2	Carboxypeptidase B2	5.46949369	−1.1090355	7.67 × 10^−6^	0.02306396
LOC107984156	ADP-Ribosylation Factor-Like Protein 17	4.86443002	−1.5030954	7.84 × 10^−6^	0.02306396
RPS25_1		4.26012384	5.18739973	9.21 × 10^−6^	0.02558409
SNAP25	Synaptosome Associated Protein 25	5.2533259	−0.8910211	1.20 × 10^−5^	0.03148975
SDR9C7	Short Chain Dehydrogenase/Reductase Family 9C Member 7	6.24842783	2.17718851	1.49 × 10^−5^	0.0373324
C8G	Complement C8 Gamma Chain	5.02875954	0.0675793	1.67 × 10^−5^	0.03967669
LINC02389	Long Intergenic Non-Protein Coding RNA 2389	5.83746521	−1.2208943	1.99 × 10^−5^	0.0441908
CPA2	Carboxypeptidase A2	4.84490407	−1.2458497	2.10 × 10^−5^	0.0441908
CST4	Cystatin S	5.06387008	−1.58686	2.12 × 10^−5^	0.0441908
TMEM78	Transmembrane Protein 78	5.8739541	0.04196411	2.42 × 10^−5^	0.04839646
LCE2B	Late Cornified Envelope 2B	5.1047095	−0.7940237	2.58 × 10^−5^	0.04970391
**WHO Criteria**					
LCE2C	Late Cornified Envelope 2C	7.4653423	1.02571863	32.0434816	8.67 × 10^−7^
KLK4	Kallikrein-related peptidase 4	6.12355248	−0.4532072	30.2043019	1.53 × 10^−6^
SLC46A2	Solute Carrier Family 46 Member 2	6.19933981	−0.4248683	30.1165666	1.58 × 10^−6^
KRT79	KRT79	7.03842812	0.45240021	28.0903087	3.01 × 10^−6^
LINC00664	Long Intergenic Non-Protein Coding RNA 664	7.17192834	1.53114803	27.9724926	3.54 × 10^−6^
RPS6KA1	Ribosomal Protein S6 Kinase A1	7.73153057	1.81676322	28.5427881	3.95 × 10^−6^
YWHAE	Tyrosine 3-Monooxygenase/Tryptophan 5-Monooxygenase Activation Protein Epsilon	4.39133048	4.33486381	26.6236527	6.23 × 10^−6^

**Table 3 pathogens-13-00510-t003:** EnrichR pathway analysis.

Software	Pathway	Adjusted *p*-Value
BioCarta (2016)	Classical Complement Pathway	0.04
BioCarta (2016)	Fibrinolysis Pathway	0.04
BioCarta (2016)	IL-7 Signal Transduction	0.04

## Data Availability

The data presented in this study are available on reasonable request from the corresponding author.

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
