# Peer review of "Differential Gene Expression in the Upper Respiratory Tract following Acute COVID-19 Infection in Ambulatory Patients That Develop Long COVID"

_pathogens, 2024, doi:10.3390/pathogens13060510_

Round 1

Reviewer 1 Report

Comments and Suggestions for Authors

The manuscript describes differences in gene expression patterns in nasopharyngeal swabs of SARS-CoV-2 infected individuals infected during the first wave of infection (presumably Wuhan variant) that display different course of the disease. Based on WHO and Thaweetha et al classifications, the patients were divided into two cohorts with classical and long COVID. Using Nextgen sequencing approach, the authors identified several series of differentially-expressed genes. These genes encoded proteins involved in blood clogging and fibrinolysis, IL-7 pathways, neurotransmission etc. The data are convincing which suggests possibility of their usage as the biomarkers, despite relatively small cohorts. The text is clearly written.

So far I have just several questions.

1. In the guidelines for Long COVID classification, several subtypes were proposed. In the current manuscript the patients were not subdivided into these groups, as is clearly seen that just few patients exhibited signs of brain fog for example (but brain fog is the main traits of one of the subroups). My question is whether it is possible to draw any conclusions which genes that were identified could be markers of this particular subgroup?

2. Can this brain fog subgroup be shown on figure 1b or analogous?

3. The font on figures 1 and 2 is too small for comprehension. Please increase it at least to 6 pt on final layout.

4. Line 270: reference is missing.

Author Response

Responses to Reviewer 1

The manuscript describes differences in gene expression patterns in nasopharyngeal swabs of SARS-CoV-2 infected individuals infected during the first wave of infection (presumably Wuhan variant) that display different course of the disease. Based on WHO and Thaweetha et al classifications, the patients were divided into two cohorts with classical and long COVID. Using Nextgen sequencing approach, the authors identified several series of differentially-expressed genes. These genes encoded proteins involved in blood clogging and fibrinolysis, IL-7 pathways, neurotransmission etc. The data are convincing which suggests possibility of their usage as the biomarkers, despite relatively small cohorts. The text is clearly written.

So far I have just several questions.

1. In the guidelines for Long COVID classification, several subtypes were proposed. In the current manuscript the patients were not subdivided into these groups, as is clearly seen that just few patients exhibited signs of brain fog for example (but brain fog is the main traits of one of the subroups). My question is whether it is possible to draw any conclusions which genes that were identified could be markers of this particular subgroup?

We agree with the reviewer that the phenotype of long-COVID is evolving to include subtypes with certain symptoms or organ system involvement and that a host-gene signature that is associated with brain fog would be useful. We have now included this in our manuscript.

2. Can this brain fog subgroup be shown on figure 1b or analogous?

This has been added to the figure.

3. The font on figures 1 and 2 is too small for comprehension. Please increase it at least to 6 pt on final layout.

We thank the reviewer for the suggestion and have enlarged the figures to make them easier to read.

4. Line 270: reference is missing.

The reference has been added

Reviewer 2 Report

Comments and Suggestions for Authors

1. RNA sequencing and bioinformatic analysis were performed on day 14, not day 7 or other time points; why? 

2. For 2.6. Statistical Analysis, the log-rank (Mantel-Cox test), and the Gehan-Breslow-Wilcoxon test were used for curve comparisons between the two cohorts (P<0.05). What are the specific outcomes for curve comparisons?

3. In the Results section, the authors claimed that lower viral loads were observed in the non-LC group throughout the course of the study. How about the positive rate at the enrollment time? 

4. When patient samples were analyzed based on the less stringent WHO criteria, fewer differentially expressed genes were identified (Figure 2b). The “less stringent” is not appropriate and should be revised. 

5. This study suggests that clinical serology may not be able to distinguish LC from non-LC patients. What is this study's original clinical or research purpose, which is to assess anti-S and anti-N serological data? 

6. Besides the smaller sample sizes, please provide other possible limitations.

Comments on the Quality of English Language

Need further English editing.

Author Response

  1. RNA sequencing and bioinformatic analysis were performed on day 14, not day 7 or other time points; why? 

As day 14 represents the end of the acute phase of COVID19 we believe that this was a potential timepoint that may provide insight into how the host immunological response may transition towards the long-COVID phenotype. We agree that analysis of additional timepoints, both earlier and later would interesting, and will be considered for future studies.

  1. For 2.6. Statistical Analysis, the log-rank (Mantel-Cox test), and the Gehan-Breslow-Wilcoxon test were used for curve comparisons between the two cohorts (P<0.05). What are the specific outcomes for curve comparisons?

These tests were used to evaluate viral load and viral RNA clearance, respectively, between the LC and non-LC patients. We have clarified this in the methods.

  1. In the Results section, the authors claimed that lower viral loads were observed in the non-LC group throughout the course of the study. How about the positive rate at the enrollment time? 

All individuals were had detectable SARS-CoV-2 RNA in their initial sample and the positivity rate was 100% at the time of enrollment. We have clarified this in the methods and results sections respectively.

  1. When patient samples were analyzed based on the less stringent WHO criteria, fewer differentially expressed genes were identified (Figure 2b). The “less stringent” is not appropriate and should be revised.

We thank the reviewer for the suggestion and have removed the term “less stringent” from the manuscript.

  1. This study suggests that clinical serology may not be able to distinguish LC from non-LC patients. What is this study's original clinical or research purpose, which is to assess anti-S and anti-N serological data?

Serological data was included to examine if this could be another potential biomarker that is predictive of the development of long COVID and could be included along with the host gene expression signature identified by RNAseq. We regret that this was unclear and have added to the manuscript to clarify this point.

  1. Besides the smaller sample sizes, please provide other possible limitations.

We appreciate the reviewer comment and have added additional limitations of our study to the discussion.

Reviewer 3 Report

Comments and Suggestions for Authors

I thank the Editor for giving me the opportunity to review the manuscript. Biondi et al. evaluated the gene expression of COVID-19 patients, in order to elucidate the host genetic profile that might pave the way to Long-COVID condition. It is a valuable aim, since long-COVID still represents a poorly known condition. However, the manuscript should be improved. Materials and methods are not clealy described, and all the analyses should be perfomerd accordingly. Limitations of the study should be highlighted and discussed properly.

Comments in details:

- Affiliation is missing for some authors.

- page 2, line 63, can you briefly describe the advantages in evaluating human RNA instead of DNA?

- page 2, "Here we identify LC patients using an established scoring criteria in a rigorously followed cohort incorporating extensive evaluation of symptoms (12) from initial diagnosis and to 90 days or greater after infection (13)": is this part of the aim? In my opinion the aim is not stated clearly enough in the text.

- page 2, paragraph 2.1: a figure or a table might help the description of inclusion/exclusion criteria from the two trials.

- the position of table 1 in the manuscript is not correct. It must be repositioned.

- page 4, line 138, "day >90": can you be more specific for this timespan?

- page 4, line 142, please edit the reference for galaxy platfrom (see https://galaxyproject.org/citing-galaxy/#primary-publication).

- figure 1, is viral load expressed as copies/mL? Please add it.

- figure(s) 2: difficult to read.

- RNAseq was performed on 27 individuals, thus all the analyses should be performed on the 27, not on the 45, which includes those without RNAseq.

- serology data should be discussed properly.

- page 10, "Although our study cohort is relatively small, the strengths include size": can you rephrase it? What are the limitions of the study?

- The fact that samples were collected from two different periods of time, which means different SARS-CoV-2 variants, should be addressed. Data about SARS-CoV-2 variants should be included in the results, if available. Different variants might elicit different gene expression patterns in the hosts.

Comments on the Quality of English Language

Minor editing of English language required.

Author Response

Responses to Reviewer 3

I thank the Editor for giving me the opportunity to review the manuscript. Biondi et al. evaluated the gene expression of COVID-19 patients, in order to elucidate the host genetic profile that might pave the way to Long-COVID condition. It is a valuable aim, since long-COVID still represents a poorly known condition. However, the manuscript should be improved. Materials and methods are not clealy described, and all the analyses should be perfomerd accordingly. Limitations of the study should be highlighted and discussed properly.

Comments in details:

1) Affiliation is missing for some authors.

Affiliations have been added.

2) page 2, line 63, can you briefly describe the advantages in evaluating human RNA instead of DNA?

We thank the reviewer for the suggestion and have added additional description to the manuscript.

3) page 2, "Here we identify LC patients using an established scoring criteria in a rigorously followed cohort incorporating extensive evaluation of symptoms (12) from initial diagnosis and to 90 days or greater after infection (13)": is this part of the aim? In my opinion the aim is not stated clearly enough in the text.

We thank the reviewer for bringing this to our attention and we have clarified this section in the introduction.

4) page 2, paragraph 2.1: a figure or a table might help the description of inclusion/exclusion criteria from the two trials.

We thank the reviewer for the suggestion yet do not think that a table would provide more information than is available in the methods. We have reworked this section of the methods to hopefully provide more clarity to the reviewers.

5) the position of table 1 in the manuscript is not correct. It must be repositioned.

The table has been repositioned

6) page 4, line 138, "day >90": can you be more specific for this timespan?

Serum samples were collected at the end of the acute phase of illness (day 14) and then during the follow up visit which took place after at least 3 months, as symptom persistence after this time is part of the criteria for identification of long-COVID cases. Collection was not performed at exactly day 90, due to challenges of booking follow up visits for the enrolled patients and thus we have listed it as “day >90”. We regret if this made anything unclear and have added more description to the methods to hopefully clarify this point.

7) page 4, line 142, please edit the reference for galaxy platform (see https://galaxyproject.org/citing-galaxy/#primary-publication).

Reference has been added

8) figure 1, is viral load expressed as copies/mL? Please add it.

We have modified the figure.

9) figure(s) 2: difficult to read.

We thank the reviewer for the suggestion and have enlarged the figures to make them easier to read.

10) RNAseq was performed on 27 individuals, thus all the analyses should be performed on the 27, not on the 45, which includes those without RNAseq.

We appreciate the perspective of the reviewer, however it is common practice to fully describe all individuals included in a trial or study in terms of demographic data, even if laboratory or other assay data is only available for a sub-set.

11) serology data should be discussed properly.

We appreciate the suggestion from the reviewer and have added a section on the serology data to the discussion.

12) page 10, "Although our study cohort is relatively small, the strengths include size": can you rephrase it? What are the limitations of the study?

We appreciate the reviewer comment and have added additional limitations of our study to the discussion.

13) The fact that samples were collected from two different periods of time, which means different SARS-CoV-2 variants, should be addressed. Data about SARS-CoV-2 variants should be included in the results, if available. Different variants might elicit different gene expression patterns in the hosts.

We thank the reviewer for their perspective. Whole genome sequencing was not widely available in our region until early 2021, and thus not all patient samples underwent this analysis as part of routine diagnosis. Additionally, not all patient samples would likely have had sufficient viral RNA to permit sequencing (as the Ct cutoff of <30 is required). As such we do not have access to the variants that all patients were infected with. Moreover, the relatively small sample size would not permit sub-analysis based on viral variant. We agree that this is an interesting avenue for future studies and have included this as a study limitation in our discussion.

Round 2

Reviewer 2 Report

Comments and Suggestions for Authors

The authors have addressed all issues.

Author Response

We thank the reviewer.

Reviewer 3 Report

Comments and Suggestions for Authors

I thank the authors for the revision. Previous concerns have been addressed and limitations have been properly highlighted.

Minor comments:

- Please review the manuscript for the presence of typos and the usage of punctuation marks.

- Lines 113-115: what is that?

- Line 259: a measure unit is missing.

- Figures and tables need a proper caption; figures and tables "should be inserted into the main text close to their first citation ". Please visit Pathogens Instructions for Authors.

Author Response

Response to reviewer 3

1) Please review the manuscript for the presence of typos and the usage of punctuation marks.

We have reviewed the manuscript for typos and punctuation.

2) Lines 113-115: what is that?

Peginterferon is a compound that contains polyethylene glycol (peg) and interferon lambda. It is an experimental treatment for COVID19. We have clarified this in the lines above and provided references.

3) Line 259: a measure unit is missing

We have corrected this in the manuscript

Figures and tables need a proper caption; figures and tables "should be inserted into the main text close to their first citation ". Please visit Pathogens Instructions for Authors.

We have modified the manuscript accordingly.